# Immunogenicity of High-Dose MVA-Based MERS Vaccine Candidate in Mice and Camels

**DOI:** 10.3390/vaccines10081330

**Published:** 2022-08-17

**Authors:** Naif Khalaf Alharbi, Fahad Aljamaan, Haya A. Aljami, Mohammed W. Alenazi, Hind Albalawi, Abdulrahman Almasoud, Fatima J. Alharthi, Esam I. Azhar, Tlili Barhoumi, Mohammad Bosaeed, Sarah C. Gilbert, Anwar M. Hashem

**Affiliations:** 1Vaccine Development Unit, King Abdullah International Medical Research Center, Riyadh 11481, Saudi Arabia; 2College of Medicine, King Saud bin Abdulaziz University for Health Sciences, Riyadh 14611, Saudi Arabia; 3Animal Facilities, King Abdullah International Medical Research Center, Riyadh 11481, Saudi Arabia; 4Department of Medical Laboratory Technology, Faculty of Applied Medical Sciences, King Abdulaziz University, Jeddah 22254, Saudi Arabia; 5Special Infectious Agents Unit, King Fahd Medical Research Center, King Abdulaziz University, Jeddah 22254, Saudi Arabia; 6Department of Medicine, King Abdulaziz Medical City, Riyadh 12746, Saudi Arabia; 7The Jenner Institute, University of Oxford, Oxford OX3 7DQ, UK; 8Department of Medical Microbiology and Parasitology, Faculty of Medicine, King Abdulaziz University, Jeddah 22254, Saudi Arabia; 9Vaccines and Immunotherapy Unit, King Fahd Medical Research Center, King Abdulaziz University, Jeddah 22254, Saudi Arabia

**Keywords:** MVA, MERS-CoV, viral vector, vaccine, antibody, camel, mice

## Abstract

The Middle East respiratory syndrome coronavirus (MERS-CoV) is a zoonotic pathogen that can transmit from dromedary camels to humans, causing severe pneumonia, with a 35% mortality rate. Vaccine candidates have been developed and tested in mice, camels, and humans. Previously, we developed a vaccine based on the modified vaccinia virus Ankara (MVA) viral vector, encoding a full-length spike protein of MERS-CoV, MVA-MERS. Here, we report the immunogenicity of high-dose MVA-MERS in prime–boost vaccinations in mice and camels. Methods: Three groups of mice were immunised with MVA wild-type (MVA-wt) and MVA-MERS (MVA-wt/MVA-MERS), MVA-MERS/MVA-wt, or MVA-MERS/MVA-MERS. Camels were immunised with two doses of PBS, MVA-wt, or MVA-MERS. Antibody (Ab) responses were evaluated using ELISA and MERS pseudovirus neutralisation assays. Results: Two high doses of MVA-MERS induced strong Ab responses in both mice and camels, including neutralising antibodies. Anti-MVA Ab responses did not affect the immune responses to the vaccine antigen (MERS-CoV spike). Conclusions: MVA-MERS vaccine, administered in a homologous prime–boost regimen, induced high levels of neutralising anti-MERS-CoV antibodies in mice and camels. This could be considered for further development and evaluation as a dromedary vaccine to reduce MERS-CoV transmission to humans.

## 1. Introduction

Modified vaccinia virus Ankara (MVA) is a non-replicating poxvirus that has been used as a viral vector for vaccines against cancer and infectious diseases [1,2]. MVA-based vaccine candidates have been developed against several pathogens, such as human immunodeficiency virus, *Plasmodium Falciparum*, hepatitis C virus, Ebola virus (EBOV), severe acute respiratory syndrome coronavirus (SARS-CoV), the Middle East respiratory syndrome coronavirus (MERS-CoV), and recently SARS-CoV-2 [3]. These vaccines have been tested in several types of animal models and in humans, with acceptable levels of safety and immunogenicity [4]. MVA has also been approved for human use as a smallpox vaccine in Europe and is stockpiled for emergency use in the USA [5,6]. In addition, there is abundant evidence in the literature showing that the safety and immunogenicity of MVA-based vaccines can be enhanced by genetic engineering, choice of promoters, and transgene insertion sites [7,8,9,10,11,12,13,14,15,16,17,18]. 

MVA infects a wide range of mammalian cells but does not replicate, leading to the activation of CD4^+^ and CD8^+^ T lymphocytes via MHC-I and MHC-II activation pathways. These pathways are induced by antigen-presenting cells, including dendritic cells (DCs). DCs are either infected by MVA or engulf proteins from MVA-infected cells. Above all, MVA is a strong stimulator of innate immune responses via the activation of TLR9 and TLR2-TLR6, MDA-5, and NALP3 inflammasome that leads to the expression of type I IFN [19,20,21]. MVA induces the expression of a large number of chemokines that induce the migration of monocytes, neutrophils, and CD4^+^ T cells to the site of inoculation (e.g., lung), which is an advantage over the use of the parental-replicating vaccinia virus (VACV) [22,23]. Therefore, MVA is an excellent choice for developing new vaccines. However, it is perceived that more than one MVA-based vaccine would be developed and approved for different pathogens in the future. The use of the same viral vector for vaccines might induce anti-vector immunity where one MVA-based vaccine might hinder the subsequent MVA-based vaccine or the second dose of the same vaccine. This was especially evident for adenoviral-based HIV vaccines where natural human pre-existing immunity hindered the vaccine efficacy [24]. 

Previously, MVA has been utilised to develop vaccines against the Middle East respiratory syndrome coronavirus (MERS-CoV) and tested in mice, camels, and humans [25,26,27,28,29,30,31]. MERS-CoV is endemic in the Middle Eastern countries and has spread to around 27 countries, causing major outbreaks in healthcare settings, mainly in South Korea and Saudi Arabia [32,33]. MERS infections have been documented in more than 2500 cases, with a 35% mortality rate in these cases. Dromedary camels have been recognised as the intermediate host for human zoonotic MERS-CoV infections. Recent data suggest that camels are 70–100% seropositive to MERS-CoV in Saudi Arabia and many Gulf Arabian and African countries, while in 2019, 35–47.5% of local camels and ~14% of imported camels in Saudi Arabia were MERS-CoV-positive based on RT-PCR [34,35,36,37]. Around half of the human primary cases of MERS reported contact with camels. In addition, neutralising antibodies and T cell responses against MERS-CoV were detected in MERS-recovered humans up to 6 years post-infection [38,39]. These lines of evidence indicate that it is plausible to develop a MERS vaccine for both camels and humans. 

Previously, we have reported the immunogenicity of an MVA-based vaccine for MERS-CoV (MVA-MERS) [25]. The vaccine incorporated a full-length spike protein of MERS-CoV and was tested in mouse models in a homologous prime–boost vaccination regimen as well as in a heterologous regimen, priming with chimpanzee adenoviral vector of Oxford University (ChAdOx1) encoding the same spike antigen of MERS-CoV and boosting with MVA-MERS. The ChAdOx1 is a potent viral vector that has been evaluated as a single-dose vaccine for MERS-CoV in mice, camels, and humans [25,30,40]. Although that heterologous regimen was superior to homologous MVA-MERS, the homologous MVA-MERS regimen induced strong antibody and T-cell responses specific to the MERS-CoV spike antigen. Here, first, we further evaluate MVA-MERS in mice, using higher dose than previously tested in which mice were given 1 × 10^7^ pfu (plaque forming unit) of MVA-MERS vaccine intramuscularly, as compared to 1 × 10^6^ in the previous study. In addition, this study allowed for evaluating the effect of anti-MVA vector immunity in the mouse model.

Second, we evaluated high-dose MVA-MERS immunisation in dromedary camels. Previously, another MVA-based MERS vaccine was tested in dromedary camels and showed that a simultaneous intramuscular and internasal immunisation, both administered as prime and then as boost, lead to high anti-MERS and anti-MVA antibody titres following that stringent regimen. Given the difficult practicality of applying this approach for camel vaccination in the field for approved vaccines, our study evaluated only intramuscular immunisation, administered in a prime–boost regimen with a high dose of 1 × 10^9^ pfu.

This study represents an investigation of increasing the doses of MVA-MERS vaccines, compared with those reported in previous studies, in experimental mice and camels and investigates vaccine anti-vector immunity in the mouse model.

## 2. Materials and Methods

### 2.1. Vaccine, Immunisation, and Samples’ Collection

MVA-MERS vaccine was constructed with the full-length spike protein from isolate (GenBank: KJ650098.1), the transgene had an upstream tPA sequence and was driven by the poxviral promoter F11, produced by The Jenner Institute, the University of Oxford as previously described [25]. In the mouse study, 3 groups of female BALB/c mice, aged 6 to 8 weeks, 10 mice per group, were intramuscularly (i.m.) immunised in the upper leg (total volume 50 µL) with a total of 1 × 10^7^ pfu of MVA; the mice were immunised in prime–boost regimens at Day 0 and Day 28 as follows: Group 1: MVA wild-type (MVA-wt) prime and MVA-MERS boost (MVA-wt/MVA-MERS); Group 2: MVA-MERS/MVA-wt; and Group 3: MVA-MERS/MVA-MERS. In the camel study, 3 groups of female dromedary camels, aged 15 to 18 months were intramuscularly (i.m.) immunised in the upper leg (total volume 2 mL) with a total of 1 × 10^9^ pfu of MVA; the camels were immunised in prime–boost regimens at Day 0 and Day 28 as follows: Group 1 (n = 2): PBS/PBS, Group 2 (n = 2): MVA-wt/MVA-wt; Group 3 (n = 4): MVA-MERS/MVA-MERS (Figure 1).

For immunisation, mice were briefly anaesthetised, using vaporised IsoFloH. Camels were physically constrained, kept in special pens, and monitored throughout the study as previously described [40]. Serum samples were collected at 0, 28, and 56 days post-immunisation (d.p.i.). In mice, 5 spleens were collected at 56 d.p.i (Figure 1). See also Appendix A for a larger and clear figure.

### 2.2. ELISA

Endpoint titres of serum antibodies were determined using an in-house indirect ELISA as previously described [41]. Briefly, 96-well Nunc ELISA plates (Thermo Scientific, Waltham, MA, USA) coated with 50 μL/well of 1 μg/mL of recombinant S1 protein (SinoBiological, Beijing, China) overnight, were washed and blocked for 1 h with 10% skimmed milk in phosphate-buffered saline (PBS) with 0.05% tween-20 (PBS-T). Then, 50 μL/well from each serum sample was added in duplicates in a 3-fold serial dilution prepared in PBS-T starting from a 1:100 dilution. After 1 h of incubation, alkaline phosphatase-conjugated goat anti-human IgG secondary antibodies (Thermo Scientific) were added at 1:3000. Finally, p-nitrophenyl phosphate substrate (pNPP) (Sigma Aldrich, St. Louis, MO, USA) was used for colour development, and the OD was measured at 405 nm. The procedure steps were all performed at room temperature. The endpoint titre for each tested serum was determined as the reciprocal value of the serum dilution that gave the OD signal converging with the cut-off determined as the average OD of a seronegative mouse or camel serum plus 3 SD, as described by previous studies [42]. For MVA whole-virus ELISA, the total protein of purified MVA-wt was quantified in Bradford Assay, 10 µg/mL was used to coat ELISA plates.

### 2.3. Mouse Ex Vivo IFN-γ ELISpot

Mouse splenocytes were harvested for analysis with IFN-γ ELISpot as previously described [43], using restimulation with 2 µg/mL of a pool of 275 peptides. These peptides are synthesised as 15 mers and overlapping by 10 amino acids, corresponding to MERS-CoV spike protein, and have been previously tested for mouse and human samples [25,30,31]. Each sample of splenocytes was plated in duplicate wells at three different numbers of cells: 500,000, 250,000, and 125,000 cells per well; thus, each sample was plated in a total of six wells. Cells were incubated with the peptide for 18–20 h at 37 °C, plates were washed, and spots were developed using a Mouse IFN-γ ELISpot kit (MabTech, Nacka Strand, Sweden), following the manufacturer’s instructions. The total number of spots in each well was counted using an AID ELISpot reader (AID Autoimmun Diagnostika, Straßberg, Germany). The numbers of spots were calculated to report the spots per 1 million splenocytes for each well. Further duplicate wells were plated with 250,000 cells from each sample and left without peptide restimulation. In the absence the restimulation, the frequencies of IFN-γ^+^ cells were subtracted from the cellular frequencies of the tested restimulated samples. IFN-γ secreting splenocytes were reported as the average of spot forming cells (SFCs) per million splenocytes for each sample.

### 2.4. MERS Pseudotyped Viral Particles (MERSpp) Neutralisation Assay

MERS pseudotyped viral particles (MERSpp) were produced in HEK293T cells and titrated using Huh7.5 cells, as previously described [44,45]. In brief, samples were prepared in a 3-fold serial dilution starting from 1:20 and added to a concentration equivalent to 200,000 RLU of MERSpp. After incubation for 1 h at 37 °C, Huh7.5 cells (10,000 cells per well) were added to the mixture. Cells only and cells with MERSpp only (both without serum) were included as controls to determine 100% and 0% neutralisation activity, respectively. cells were lysed after 48 h, and the assay was developed using Bright-Glo™ Luciferase Assay System (Promega). IC_50_ neutralisation titres were calculated for each serum sample using the GraphPad Prism software.

### 2.5. Data Analysis

All raw data were reading signals collected from ELISA microplate readers for antibody responses; an ELISpot counter for mouse ex vivo IFN-γ ELISpot assays; or a luminometer for luciferase lights as readouts of the MERSpp neutralisation assay. Data were collected in a Microsoft Excel spreadsheet and were placed and analysed using the GraphPad Prism software, Version 9.3.1. (GraphPad Software, Inc., San Diego, CA, USA) This software was used to perform all statistical analyses.

## 3. Results

### 3.1. MVA-MERS Immunogenicity in Mouse Models

Serum samples from three groups of mice, namely Group 1: MVA-wt/MVA-MERS; Group 2: MVA-MERS/ MVA-wt; and Group 3: MVA-MERS/MVA-MERS, were evaluated at 0, 28, and 56 d.p.i for humoral immune responses. Strong anti-MERS spike antibody (Ab) responses were induced at 28 d.p.i., post-prime, in Groups 2 and 3. The level of Abs was enhanced after the boost (56 d.p.i) in Group 3. In Group 2, in which mice were primed with MVA-MERS and boosted with MVA-wt, no significant change in the level of Ab responses was observed. However, comparing Groups 1 to 3, primed with MVA-wt and MVA-MERS, respectively, and boosted both with MVA-MERS, the anti-MERS-CoV spike Ab responses increased in both groups. Although using MVA with the same antigen (spike protein) for prime–boost enhanced the antigen-specific Ab responses (Group 3), priming with MVA-wt and boosting with MVA-MERS in Group 1 showed a high increase in S-specific Ab responses (Figure 2A). Comparing antibody responses between Group 1 at day 56 and Group 3 at day 28, Ab responses to MVA-MERS were similar in mice that had received a dose of MVA-wt and those that had not. The second dose in Group 1 was indeed a priming dose with MVA-MERS, which indicates that anti-vector (MVA) immune responses may not hamper the immune responses for a subsequent vaccine with a different antigen (Figure 2B). In contrast, the boosting mice that received MVA-MERS with MVA-wt did not impact S-specific immune Ab responses for Group 2. The lower increase in Ab levels in this group could be mainly due to the longer duration between MVA-MERS and sampling at 56 d.p.i. (Figure 2B). The neutralisation activities of induced antibodies were confirmed for all the samples, with similar results as those observed in the binding Abs (Figure 2C). See also Appendix A for a larger and clear figure.

On Day 56 post-immunisation, the IFN-γ-producing T cells, as measured by spleen ex vivo ELISpot, showed strong responses in the MVA-MERS prime–boost regimen (Group 3). There was an insignificant increase in the IFN-γ T-cell responses in Group 2, compared with Group 1, but this is likely due to the timing of giving MVA-MERS; it was administered at 28 d.p.i for Group 1 and at 0 d.p.i for Group 2. This means there was a one-month difference in assessing the T-cell immune responses in Groups 1 and 2 (Figure 2D). Nonetheless, these data indicate that two doses of MVA-MERS elicit stronger S-specific humoral and cellular responses in mice.

### 3.2. MVA-MERS Immunogenicity in Dromedary Camels

Three groups of MERS-seropositive young camels were immunised in homologous prime–boost regimens, namely Group 1: PBS; Group 2: MVA-wt; and Group 3: MVA-MERS. Serum and nasal swab samples were evaluated at 0, 28, and 56 d.p.i for humoral immune responses. High levels of anti-MERS spike antibody (Ab) responses were observed at 0 d.p.i (pre-immunisation), indicating the level of MERS-CoV seroprevalence in camels. At 28 d.p.i, there was no difference between immunised groups. However, following the boost, at 56 d.p.i, the MVA-MERS vaccine induced higher levels of the anti-MERS-CoV spike Abs (Figure 3A). The neutralisation activity of anti-MERS-CoV camel Ab responses was confirmed. Although there was a distinguishing difference between MVA-wt (Group 2) and MVA-MERS (Group 3) at 28 d.p.i. (After one dose of the vaccine), the neutralising Abs were at similar levels at 56 d.p.i. (Figure 3B). The nasal swab testing showed that the neutralising Abs against MERS-CoV pseudoviruses in nasal secretion were at low levels, with no differences between groups (Figure 3C). See also Appendix A for a larger and clear figure.

### 3.3. Anti-MVA Immune Responses in Vaccinated Mice and Camels

Sera from vaccinated mice and camels were utilised to evaluate antibody responses to the vaccine vector, MVA. Regardless of the vaccine-encoded antigen, anti-MVA antibody responses to two doses of MVA were similar in all the groups of either mice or camels (Figure 4A,B). See also Appendix A for a larger and clear figure. This indicates that anti-MVA antibodies were not affected by the vaccine antigen or the lack of the vaccine antigen (in MVA-wt). Furthermore, these results confirm that the presence of anti-MVA Abs does not hinder the production of antigen-specific immunity in mice and camels.

## 4. Discussion

In this study, an MVA-based vaccine for MERS-CoV, incorporating a full-length spike antigen was tested in a mouse model and in dromedary camels. The vaccine induced high levels of binding and neutralising Abs against the spike, with strong T-cell-mediated immune responses in mice, and high levels of binding antibodies in camels. In mice, the homologous prime–boost vaccination regimen provides evidence that MVA is an efficient viral vector for priming and boosting. The use of MVA-based vaccine in prime–boost immunisation has been shown to be sub-optimal, compared with the use of heterologous prime–boost regimens in which, for example, priming with ChAdOx1-based vaccine and boosting with MVA encoding the same vaccine induced higher neutralising Abs in mouse models [25]. This led to the development of vaccines based on two different vectors or platforms in order to circumvent the immune responses specific to the priming vector. In addition, pre-existing immune responses to a vaccine vector have been shown to hamper the vector-based vaccine. This was reported for the use of human adenovirus as a viral vector for HIV vaccine candidates [46]. However, in the almost complete lack of anti-poxvirus (MVA) immunity in humans, due to the eradication of smallpox, the MVA-based vaccine may not be affected by pre-existing immunity, but there could still be concerns over the use of MVA as a vector for multiple antigens or multiple vaccines, if any approved in the future. 

Our study attempted to address these concerns and reported that anti-MVA Ab responses post-prime were similar to post-boost levels, in both types of animals. This indicates that anti-MVA-induced immune responses at prime did not affect the anti-MERS-CoV spike Ab level, which was boosted to a higher level. MVA-MERS in mice that had MVA-wt induced high levels of anti-spike Ab responses, which indicates that MVA has a strong effect in priming immune responses in animals that already had anti-MVA antibodies. In homologous vaccination regimens of MVA vaccines, the increases in anti-antigen Ab levels are usually lower post-boost than post-prime, which is supported by the current data and previous reports. 

Most dromedary camels in Saudi Arabia and part of Middle Eastern and African countries are MERS-CoV-seropositive [34,47,48,49,50,51,52,53]. Our previous study supported the idea of MERS vaccination of seropositive camels to achieve partial or complete protection [40]. Here, seropositive camels were utilised for a homologous prime–boost vaccination with an MVA-MERS vaccine candidate. Post-prime, MVA did not induce an evident increase in anti-MERS-CoV spike Abs. Post-boost, the vaccine candidate showed strong levels of binding and neutralising Abs. However, the neutralising Abs at 56 d.p.i were similar in both groups of camels, unlike the binding antibodies. This weak response of neutralising Ab levels with an insignificant difference between MVA-MERS-vaccinated camels and MVA-wt-immunised camels at Day 56 is interesting and requires further studies. It could be, however, due to assay differences. Previously, we have reported fewer differences in the Ab levels when using pseudoviral neutralisation assay than when using ELISA, which are different assays with different readout mechanisms and, therefore, may not ensure the reproducibility of one assay’s results by the other. In addition, a previous study showed that priming with an MVA-based MERS vaccine candidate, administered intranasally and intramuscularly, did not induce detectable neutralising Abs in all seronegative camels [29]. Although we attempted to increase the dose, administered only intramuscularly, our finding suggests there is still room to optimise the camel MERS vaccine to be more efficient, with fewer application difficulties.

The MERS-CoV S protein has been the focus for most MERS-CoV vaccine candidates published to date. Several vaccine candidates based on full-length or truncated S protein including virally-vectored, DNA-based, nanoparticle-based or protein-based subunit vaccines have been developed and investigated in experimental animal models and humans by several groups including our group [25,31,40,54,55,56,57,58,59,60,61,62,63]. Two virally vectored vaccines and one DNA-based vaccine have reached phase I clinical trials [30,31,60,61]. Furthermore, we and others have investigated the efficacy of different MERS vaccine candidates including three virally vectored and one DNA-based vaccines in camels experimentally or in field studies [29,40,57,62]. In our previous report, a single dose of ChAdOx1 MERS vaccine showed partial but highly significant protection in seropositive camels over 3 weeks follow-up period. This protected immunised camels from a natural infection when co-housed with PCR-positive virus-shedding camels. Only 3 out of 5 camels showed transient infection (detected by PCR) for 1 to 3 days post challenge [40]. Another study that used 2-doses of MVA-based MERS vaccine, each dose as simultaneous intranasal/intramuscular, have shown that the seronagative camels had significant reduction in infectious virus and viral RNA transcripts after MERS-CoV experimental infections [29]. On the other hand, the other two studies mainly focused on the immunogenicity without any challenge studies in camels [57,62]. Although we were not able to conduct a challenge study here, we attempted to increase the dose, given only intramuscularly, and our finding suggests there is still a room to optimise camel MERS vaccine to be more efficient with less application difficulties. In mice, most studies have shown that single dose of ChAdOx1 MERS or 2-doses of MVA MERS vaccines showed complete protection in transgenic or transduced mouse models [27,58].

This study has some limitations, including its lack of a challenge model to evaluate the efficacy of the vaccine. All camels were seropositive for MVA, which requires further analysis. Since these were juvenile camels, it is unlikely that they were exposed to a poxvirus such as camelpox, but they may have maternal anti-poxvirus from exposed mother camels. Since MVA has around 200 proteins, it is also possible that these camels have some Abs for other pathogens that were cross-reactive. Further experiments are required to set up an MVA (or poxvirus)-specific ELISA rather than a whole-virus ELISA.

In the lack of a high number of infections in humans, a MERS vaccine in humans may not be properly developed or evaluated. Therefore, MERS-CoV vaccines for camels are believed to serve as a one-health intervention where a vaccine for camels could reduce the virus circulation in camels in endemic countries such as Saudi Arabia, leading to a reduction in spill-over infections to humans. This study builds on the growing evidence that camel MERS vaccination could be achieved using an MVA viral vector. 

## 5. Conclusions

An MVA-based MERS vaccine was administered to mice and induced strong neutralising Abs and T-cell responses. The vaccine induced strong neutralising Abs in juvenile camels. This vaccine can be further developed to evaluate its durable immunogenicity and efficacy as a one-health vaccine for MERS-CoV.

## Figures and Tables

**Figure 1 vaccines-10-01330-f001:**
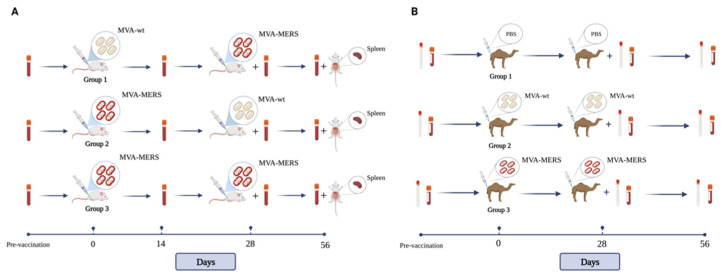
Diagram representation of mouse (**A**) and camel (**B**) vaccination studies. See also Appendix A for a larger and clear figure.

**Figure 2 vaccines-10-01330-f002:**
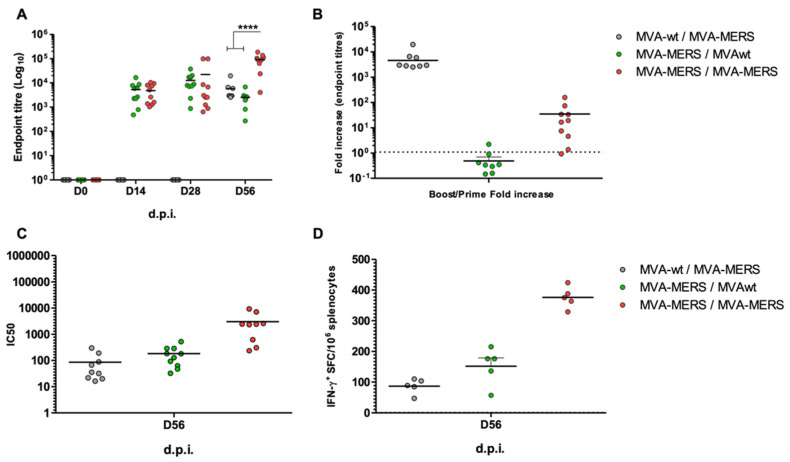
**Humoral and cellular immunogenicity of MVA-wt and MVA-MERS prime–boost vaccination in BALB/c mice.** Balb/c mice (n = 10 per group) were intramuscularly immunised with 10^7^ PFU as follows: Group 1: MVA-wt/MVA-MERS (grey symbols); Group 2: MVA-MERS/MVA-wt (green symbols); and Group 3: MVA-MERS/MVA-MERS (red symbols). Serum samples were collected pre-vaccination at 0 d.p.i, at 28 d.p.i (post-prime), and 56 d.p.i (post-boost). Anti-MERS-CoV spike IgG antibodies were measured using ELISA at all time points, reported as Log10 endpoint titres (**A**). The fold increases in IgG Ab endpoint titres from the prime time (28 d.p.i) to the boost time (56 d.p.i) are presented (**B**). Neutralisation activity of serum antibodies at 56 d.p.i. was confirmed via MERSpp neutralisation assay (**C**), presented as the serum dilution that showed inhibitory concentration of 50% of the pseudoviruses (IC50). At 56 d.p.i, IFN-γ ex vivo ELISpot was performed on splenocytes, collected at 56 d.p.i (**D**). Mean values are shown as lines. **** indicates *p* value < 0.0001 by one-way ANOVA test and Bonferroni’s multiple-comparison post-test.

**Figure 3 vaccines-10-01330-f003:**
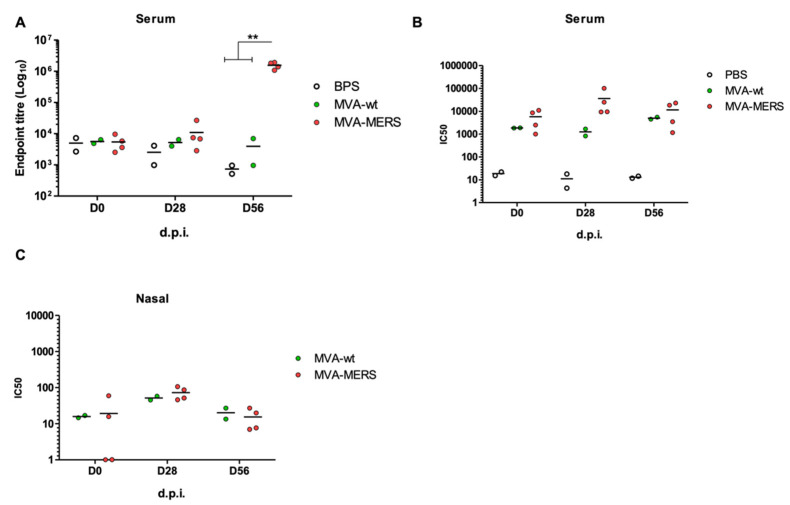
**Humoral immunogenicity of homologous MVA-MERS prime–boost vaccination in dromedary camels.** Camels (n = 4) were intramuscularly immunised with 1 × 10^9^ PFU. Other camels were immunised with similar dose of MVA-wt (n = 2) and PBS (n = 2). Serum samples were collected pre-vaccination at 0 d.p.i, at 28 d.p.i (post-prime), and 56 d.p.i (post-boost). Anti-MERS-CoV spike antibodies were measured with ELISA at all time points, reported as Log10 endpoint titres (**A**). Neutralisation activity of antibodies was confirmed via MERSpp neutralisation assay for serum (**B**) and nasal swab (**C**) samples collected at all time points, presented as the sample dilution that showed inhibitory concentration of 50% of the pseudoviruses (IC50). Mean values are shown as lines. ** indicates *p* value < 0.002 by one-way ANOVA test and Bonferroni’s multiple-comparison post-test.

**Figure 4 vaccines-10-01330-f004:**
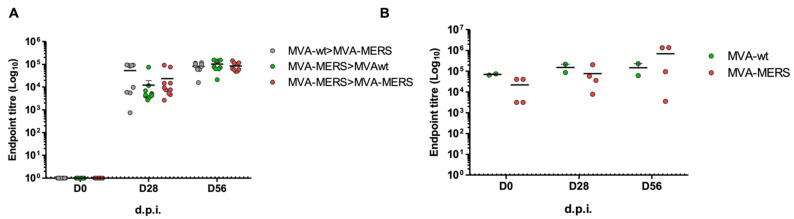
**Antibody specific to MVA in immunised mice and camels.** Serum samples collected from immunised mice (**A**) and camels (**B**), as explained in Figure 2 and Figure 3, were tested in a whole-virus in-house ELISA for anti-MVA antibodies, reported as Log10 endpoint titres. Mean values are shown as lines.

## Data Availability

Interested investigators should contact Naif K. Alharbi for information on data sharing.

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
