# Peer review of "Immunogenicity of High-Dose MVA-Based MERS Vaccine Candidate in Mice and Camels"

_vaccines, 2022, doi:10.3390/vaccines10081330_

Round 1
Reviewer 1 Report
Manuscript ID: vaccines-1794051
Title: Immunogenicity of High-dose MVA-based MERS- Vaccine Candidate in Mice and Camels
Authors: Naif Khalaf Alharbi *, Fahad Aljamaan, Haya A. Aljami, Mohamed W.
Alenazi, Abdulrahman Almasoud, Hind Albalawi, Fatima J. Alharthi, Esam I Azhar, Tlili Barhoumi, Mohammad Bosaeed, Sarah C. Gilbert, Anwar M. Hashem
Submitted to section: Veterinary Vaccines
This study investigates increasing doses of an MVA-based vaccine for MERS-CoV in mice and camels. It also investigates anti- vaccine vector antibodies in the mouse and the camel.
The authors say it represents the prosecution of a work published by the Authors, but the paper they quote investigates ChAdOx1 MERS. They should revise this part of the intro.
One objection is
What is generally done is test on mice then on the relevant spesies and this study goes the other way round.
Secondly, the use of immune camels in the study needs to be explained. I believe it would have been better to use naïve animals.
In addition, positive control sera from animals and/or human that have had the disease could be added as a positive control, for example the ones used in Ref. 39.
Minor comments
Lines 199-200. With no differences between groups of previously infected animals, how can the Authors be sure the antibodies are due to immunization?
Figures. Generally speaking, statistical analysis is missing from all figures.
Axes should state endpoint titer of what, IC50 of what, fold increase of what and compared to what?
Figure 1 cMVA-MERS is a different acronym in the paper, it is confusing. Please use the same.
In addition: Please make it clear in the figure that the first blood sample was drawn at day 0.
B: how do the authors explain that higher increases are obtained with wt+MERS MVA than with MERS-MERS?
Fig. 3 letter C missing. Most importantly, there does not seem to be much difference in anti SARS AB between camels immunized with MVA-wt and MVA-MERS
Fig. 4 1:1000 Ab response against MVA at day 0 in sera from mice (A). Any explanation?
C is not defined.
My conclusions do not match those by the Authors:
Antibody immunity is not specific in camels immunized with MVA-MERS (Fig4B) and very similar to that of animals immunized with wt MTA by day 56. It does not seem to be “strong”, especially with no positive control to compare to.
Anti MTA vector Ab seem to be elicited significantly in camels but not in mice (Fig 4 B vs A), which is surprising.
Nasal immunity can only be seen at day 28 in camels, but it is very similar to that of animals immunized with wt MT.
Author Response
Point-by-point Responses to Reviewer 1 Comments:
Reviewer 1
Comments and Suggestions for Authors
Manuscript ID: vaccines-1794051
This study investigates increasing doses of an MVA-based vaccine for MERS-CoV in mice and camels. It also investigates anti- vaccine vector antibodies in the mouse and the camel.
- We thank the Reviewer for their time and review of our manuscript.
The authors say it represents the prosecution of a work published by the Authors, but the paper they quote investigates ChAdOx1 MERS. They should revise this part of the intro.
- The cited reference was a mistake; we have now updated the cited reference to our previous MVA-MERS paper (Alharbi NK, Padron-Regalado E, Thompson CP, et al. ChAdOx1 and MVA based vaccine candidates against MERS-CoV elicit neutralising antibodies and cellular immune responses in mice. Vaccine. 2017;35(30):3780-3788. doi:10.1016/j.vaccine.2017.05.032).
One objection is
What is generally done is test on mice then on the relevant spesies and this study goes the other way round.
- We have tested MVA-MERS in mice, in a previous study, cited above; also this study started by a mouse study then moved into a camel study.
Secondly, the use of immune camels in the study needs to be explained. I believe it would have been better to use naïve animals.
- We thank the Reviewer for this point. Previously, we have reported that around 90% of camels are seropositive for MERS-CoV; in addition, literature shows that 70-100% of camels are MERS seropositive, in the Middle East and Africa. We have also previously reported that a vaccine candidate showed a protection only in seropositive camels; thus this study utilized seropositive camels. These points are now stated in the manuscript.
In addition, positive control sera from animals and/or human that have had the disease could be added as a positive control, for example the ones used in Ref. 39.
- We thank the Reviewer for this point. We have included some positive and negative sera from previously collected camels as internal controls for the ELISA test. These are not shown in the figures because we report the endpoint titre, which calculates the OD values of the sample dilution that converge with the OD value of the negative sera, plus 3xSD.
Minor comments
Lines 199-200. With no differences between groups of previously infected animals, how can the Authors be sure the antibodies are due to immunization?
- We agree with the Reviewer and have corrected this in the manuscript.
Figures. Generally speaking, statistical analysis is missing from all figures.
- Statistical analysis has now been performed on all figures (data). The statistical significant differences are shown by asterisk on figures and explained in figure legends.
Axes should state endpoint titer of what, IC50 of what, fold increase of what and compared to what?
- We thank the reviewer for this suggestion; we have now elaborated and explained the axes in the figure legends.
Figure 1 cMVA-MERS is a different acronym in the paper, it is confusing. Please use the same. In addition: Please make it clear in the figure that the first blood sample was drawn at day 0.
- We apologise for this mistake and thank you for the suggestions; figure 1 has now been amended accordingly and improved to show the “pre-vaccination” sampling.
B: how do the authors explain that higher increases are obtained with wt+MERS MVA than with MERS-MERS?
- We apologise for any confusion here; Figure 2B shows the increase in Anti-MERS spike Ab after the boost, at 56 d.p.i, by this time all mice, all groups, had received the vaccine either at prime or at boost (or at both prime and boost). We clarified the figure labels that are shown within the figure itself to make it clearer.
Fig. 3 letter C missing. Most importantly, there does not seem to be much difference in anti SARS AB between camels immunized with MVA-wt and MVA-MERS
- The letter C for the 3rd graph of figure 3 has now been defined in the legend. This graph shows anti-MERS spike neutralizing Ab in nasal swab with no difference between MVA-wt and MVA-MERS, we agree with the reviewer and have amended the text (line 199-200) accordingly.
Fig. 4 1:1000 Ab response against MVA at day 0 in sera from mice (A). Any explanation?
C is not defined.
- It seems the figure 4 we have placed within the manuscript is a replica of figure 3. The figure 4 uploaded in the journal system is different than that in the manuscript. Sincere apology for this hastiness and mistake!
- We have now placed figure 4 within the manuscript. For figure 4, mice sera Ab presented in graph A shows 0 Ab responses at day 0.
- There is no graph C in the newly inserted figure 4.
My conclusions do not match those by the Authors:
Antibody immunity is not specific in camels immunized with MVA-MERS (Fig4B) and very similar to that of animals immunized with wt MTA by day 56. It does not seem to be “strong”, especially with no positive control to compare to.
- As in our response above, we really sincerely apologise for the mistake of not adding Figure 4 in the paper. Figure 3 was copied twice instead of Figure 3. The revised manuscript now has the actual figure 4. We would be happy to receive the Reviewers’ comments on this figure.
Anti MTA vector Ab seem to be elicited significantly in camels but not in mice (Fig 4 B vs A), which is surprising.
- Apology again for the confusion on figure 4. We are happy to receive the reviewer’s further comments on the newly inserted figure 4, in the revised manuscript.
Nasal immunity can only be seen at day 28 in camels, but it is very similar to that of animals immunized with wt MT.
- We agree on this and have amended the manuscript accordingly (Lines: 199-200)
Reviewer 2 Report
A modified Vaccinia virus Ankara (MVA) viral vector encoding a full-length spike protein of MERS-CoV was tested for the immunogenicity of high-dose MVA-MERS in prime-boost vaccinations in mice and camels. The result showed that Two high-doses of MVA-MERS induced strong Ab responses in both mice and camels including neutralising antibodies. This study has a good effect on the development of MERS vaccine, but it also has some needs to be improved .
1. ChAdOx1-based MERS-CoV 84 vaccine should be introduced in detail.
2. Why did the single-dose and double-dose studies in this study compare together? Because there could be different results from batch to batch, right?
3. Group 1 (n=2): 115 PBS/PBS, Group 1 (n=2): MVA-wt/ MVA-wt, Group 3 (n=4): MVA-MERS/MVA-MERS. Two camels per group were not statistically significant.
4. Data analysis should be introduced in detail.
5. MVA-wt/MVA-MERS must be marked clearly in the figure. Wt >MERS is not omitted in Fig.2.
6. “3.2 MVA-MERS immunogenicity in dromedary camels”What are you trying to clarify?
Author Response
Point-by-point Responses to Reviewer 2 Comments:
Reviewer 2
Comments and Suggestions for Authors
A modified Vaccinia virus Ankara (MVA) viral vector encoding a full-length spike protein of MERS-CoV was tested for the immunogenicity of high-dose MVA-MERS in prime-boost vaccinations in mice and camels. The result showed that Two high-doses of MVA-MERS induced strong Ab responses in both mice and camels including neutralising antibodies. This study has a good effect on the development of MERS vaccine, but it also has some needs to be improved .
- We thank the Reviewer for their time and review of our manuscript.
- ChAdOx1-based MERS-CoV 84 vaccine should be introduced in detail.
- We thought the scope of the paper is more focused on MVA-MERS rather than MERS vaccines, in general. However, we have now elaborated more on the ChAdOx1 and cited references for further reading.
- Why did the single-dose and double-dose studies in this study compare together? Because there could be different results from batch to batch, right?
- The MVA-MERS vaccine was produced as one batch for both the mice and camel studies. Different vials were thawed prior to immunization of mice and camels, but from the same batch. Both in mice and in camels, we compared 2 doses of MVA-MERS to 2 doses of MVA-wt (in camels) or 2 heterologous doses of MVA-MERS and MVA-wt (in mice). Single dose regimens were not studied, neither in mice or in camels.
- Group 1 (n=2): 115 PBS/PBS, Group 1 (n=2): MVA-wt/ MVA-wt, Group 3 (n=4): MVA-MERS/MVA-MERS. Two camels per group were not statistically significant.
- The camel experiment has been conducted in a remote farm and it has huge logistical difficulties to be conducted by an academic institution. We reviewed the literature and found a study on another MERS vaccine that has been conducted on as small number as our study (Haagmans, B. et al). We agree with the reviewer that statistical power may not be applicable on this sample size, but we worked to present the scientific finding and provide more data on these rarely-used animals for vaccine immunogenicity. Therefore, we are hopeful that the Editor and Reviewers could consider our manuscript for publication.
- Data analysis should be introduced in detail.
- This has now been added.
- MVA-wt/MVA-MERS must be marked clearly in the figure. Wt >MERS is not omittedin Fig.2.
- This has now been corrected.
- “3.2 MVA-MERS immunogenicity in dromedary camels”What are you trying to clarify?
- This study was conducted to evaluate the dromedary camel immune responses elicited by MVA-MERS vaccine candidate that has been designed and produced by the team in the Jenner institute. As stated in the manuscript introduction, this vaccine candidate was studied in mouse models in homologous and heterologous prime-boost vaccination but was never tested in camels. This study also showed that high doses of MVA were able to increase Ab levels in seropositive camels after 2-doses. There was only one study on MVA based MERS vaccines in camels, by Haagmans et al, that utilized sero-negative camels, but the majority of camel population in the Middle East and Africa is MERS-CoV sero-positive already.
- We would be happy to receive and address any specific comments/request from the Reviewer.
Reviewer 3 Report
Alharbi and colleagues report on the development of an MVA-based vaccine against MERS-CoV and evaluate the immunogenicity of the vaccine in both mice and camels. As the recent COVID-19 pandemic has shown us, we are constantly at risk of zoonotic coronaviruses making the jump to humans, and while the loss of life due to SARS-CoV-2 is tragic, we were fortunate that it was not a virus with a mortality rate like MERS-CoV. Having a vaccine against MERS-CoV is an important public health benefit, and the ability to immunize animal reservoirs such as camels presents an opportunity. The authors show that the MVA-MERS vaccine is immunogenic in mice, but arguably less so in camels. The authors also demonstrate that the use of the MVA vector in the vaccine schedule does not compromise immune responses to subsequent immunizations, which is beneficial if this platform is to be developed for this or other vaccines. While the work is scientifically sound, I have several issues that the authors will need to address and revise before this article can be considered for publication.
Comments:
Abstract, Line 23: Change “could” to “can” since we know that MERS-CoV can infect humans.
Abstract, Line 24: Change “death confirmed cases” to “mortality rate” for more concise language.
Introduction, Line 43, 44: Abbreviations should be written out when first used, e.g. hepatitis C virus (HCV), etc.
Introduction, Line 43: P. falciparum should be italicized.
Methods 2.3, Line 138: The method for the ELISpot is not well described. How many replicate wells were used? How many cells were plated per well? For how long were the cells stimulated? Which positive and negative controls were used? This information should be added here, even if the method is referenced from another publication.
Methods 2.5, Line 157: The software version for GraphPad Prism should be provided.
Results 3: Whenever a figure is referenced in the text, the authors should place the figure name in parentheses rather than adding it after a comma, e.g. from Line 171 “group 1 showed high increase 170 in S-specific Ab responses (Figure 2A).” This should be done for the entire manuscript wherever a figure is referenced.
Results, Line 184: The authors indicate that there is a one month difference in the T cell assessment for MERS-CoV response between groups 1 and 2 based on the immunization schedule. Why not include enough mice to also collect spleens from group 2 at Day 28 so that the responses for groups 1 and 2 could be compared after equivalent time periods after the prime as well as the Day 56 time point?
Results 3.2, Line 189: Do you suspect that existing MERS-CoV antibodies in the seropositive camels may have impacted the immune response to the vaccine, and that may explain why the response was weaker?
Results 3.2, Line 198: Somewhat concerning that the IgG titer increased considerably after the boost immunization in camels, but when looking at Figure 3, the neutralization titer seems to have decreased for all 4 animals. This would suggest that the antibody response is being focused towards non-protective epitopes perhaps, but a sample size of n=4 is hard to draw conclusions from at this stage. The authors should address this observation in their discussion.
Results 3.2, Line 201: Did you look at IgA response in the nasal mucosa of the immunized camels?
Discussion, Line 236: Should read “lower post-boost”
Figure 2. I would change the inset figure legend to have the “/” separating the vaccine doses. Using the “>” could mislead the reader since it is commonly used in mathematics.
Figure 2, Panel B is slightly misleading. Is this fold increase between the prime and boost? If so, the wt/MERS response is being compared against zero whereas the MVA-MERS/MVA-MERS group has a titer at Day 28, so I don’t think it’s directly relevant to compare those. Maybe compare all groups at D56 to D0, and for MVA-MERS/MVA-MERS you can show the fold increase from D28 to D56 as a separate grouping.
Figure 4: This appears to be the same figure as Figure 3 and does not match with the Figure legend. Please revise to include the appropriate figure panels.
Author Response
Point-by-point Responses to Reviewer 3 Comments:
Reviewer 3
Comments and Suggestions for Authors
Alharbi and colleagues report on the development of an MVA-based vaccine against MERS-CoV and evaluate the immunogenicity of the vaccine in both mice and camels. As the recent COVID-19 pandemic has shown us, we are constantly at risk of zoonotic coronaviruses making the jump to humans, and while the loss of life due to SARS-CoV-2 is tragic, we were fortunate that it was not a virus with a mortality rate like MERS-CoV. Having a vaccine against MERS-CoV is an important public health benefit, and the ability to immunize animal reservoirs such as camels presents an opportunity. The authors show that the MVA-MERS vaccine is immunogenic in mice, but arguably less so in camels. The authors also demonstrate that the use of the MVA vector in the vaccine schedule does not compromise immune responses to subsequent immunizations, which is beneficial if this platform is to be developed for this or other vaccines. While the work is scientifically sound, I have several issues that the authors will need to address and revise before this article can be considered for publication.
- We thank the Reviewer for their time and review of our manuscript.
Comments:
Abstract, Line 23: Change “could” to “can” since we know that MERS-CoV can infect humans.
Abstract, Line 24: Change “death confirmed cases” to “mortality rate” for more concise language.
Introduction, Line 43, 44: Abbreviations should be written out when first used, e.g. hepatitis C virus (HCV), etc.
Introduction, Line 43: P. falciparum should be italicized.
- The above have now been amended as suggested.
Methods 2.3, Line 138: The method for the ELISpot is not well described. How many replicate wells were used? How many cells were plated per well? For how long were the cells stimulated? Which positive and negative controls were used? This information should be added here, even if the method is referenced from another publication.
- ELISpot section has been elaborated on with details; we are happy to address any further suggestion from the Reviewer.
Methods 2.5, Line 157: The software version for GraphPad Prism should be provided.
- This has now been dded.
Results 3: Whenever a figure is referenced in the text, the authors should place the figure name in parentheses rather than adding it after a comma, e.g. from Line 171 “group 1 showed high increase 170 in S-specific Ab responses (Figure 2A).” This should be done for the entire manuscript wherever a figure is referenced.
- This has now been amended throughout the manuscript, as suggested.
Results, Line 184: The authors indicate that there is a one month difference in the T cell assessment for MERS-CoV response between groups 1 and 2 based on the immunization schedule. Why not include enough mice to also collect spleens from group 2 at Day 28 so that the responses for groups 1 and 2 could be compared after equivalent time periods after the prime as well as the Day 56 time point?
- In that paragraph of the results, we wanted to highlight that there was a timing difference between group 1 and 2, to alert readers to be cautious when comparing the data. However, the values of ELISpot are overlapping and not significantly different, even with this timing difference, there is only a trend increase likely due to the timing. It is not possible for us to conduct a new mouse study, given the time and funding constrains; and we believe the main message of figure 4D is to show that 2-doses of MVA-MERS induced stronger T cell responses than heterologous MVA-MERS> MVA-wt or MVA-wt>MVA-MERS regimens.
Results 3.2, Line 189: Do you suspect that existing MERS-CoV antibodies in the seropositive camels may have impacted the immune response to the vaccine, and that may explain why the response was weaker?
- The Ab responses in camels were higher in MVA-MERS vaccinated camels as compared to MVA-wt or PBS (in ELISA, Figure 3A); this confirms that the vaccine boosted the pre-existing immune responses with no clear impact from the pre-existing immunity. In addition, the vaccine is not a whole-inactivated virus, so pre-existing Ab should not react to the vaccine, which is an MVA that encodes MERS-CoV spike protein.
- The difference in NAb levels between MVA-MERS vaccinated camels and MVA-wt immunized camels at day 56 (Figure 3B) was small, unlike the binding Ab in ELISA. This might be a results of differences in the used assays. Previously, we have seen less differences in the Ab levels when looking at pseudoviral neutralization assay results as compared to ELISA results. They are completely different assays and different readout mechanisms that may not ensure reproducibility of one assay’s results by another assay.
- Nevertheless, we have included this point in our revised discussion and we remain happy to address any suggestions or comments from the kind Reviewer.
Results 3.2, Line 198: Somewhat concerning that the IgG titer increased considerably after the boost immunization in camels, but when looking at Figure 3, the neutralization titer seems to have decreased for all 4 animals. This would suggest that the antibody response is being focused towards non-protective epitopes perhaps, but a sample size of n=4 is hard to draw conclusions from at this stage. The authors should address this observation in their discussion.
- We thank the Reviewer for this important point, we have now addressed this point in the Discussion.
Results 3.2, Line 201: Did you look at IgA response in the nasal mucosa of the immunized camels?
- We have not measured IgA per se. The nasal samples were tested in the pseudoviral neutralization assay that cannot distinguish the Ig type.
Discussion, Line 236: Should read “lower post-boost”
Figure 2. I would change the inset figure legend to have the “/” separating the vaccine doses. Using the “>” could mislead the reader since it is commonly used in mathematics.
Figure 2, Panel B is slightly misleading. Is this fold increase between the prime and boost? If so, the wt/MERS response is being compared against zero whereas the MVA-MERS/MVA-MERS group has a titer at Day 28, so I don’t think it’s directly relevant to compare those. Maybe compare all groups at D56 to D0, and for MVA-MERS/MVA-MERS you can show the fold increase from D28 to D56 as a separate grouping.
- More details have been added in the figure legends.
Figure 4: This appears to be the same figure as Figure 3 and does not match with the Figure legend. Please revise to include the appropriate figure panels.
- We apologise for this mistake; figure 4 has now been added.